# Plant Phenotyping from Limited Training Data by Active Learning and Annotation

## Abstract

AI-based plant phenotyping has significantly expanded the scope, scale, and speed of trait data collection. Phenotyping of epidermal cell patterning is a long-standing target of studies on carbon and water relations due to the important functions of stomata, bulliform cells and veins. Although SD estimation has benefited from AI-based techniques, automated segmentation and analysis of costal zones and bulliform cell regions have received much less attention. One major bottleneck is that AI tools often require manual annotation of large datasets for training, which is labor-intensive and requires domain expertise. In addition, models trained on one species frequently perform poorly on others. In this paper, we propose an automated framework that enables the detection of stomata, costal zones, and bulliform cell regions, leveraging domain knowledge and minimizing the need for extensive training data. We extract features from both topographic and intensity data, separating them using knowledge of spatial structure, specifically the repeatable, linear, periodic arrangement of epidermal cells, and intrinsic cell models to minimize noise. We bootstrap learning by starting with the most structured parts of the images and progressively adding less structured regions. Furthermore, we incorporate active annotation to continually expand the training dataset throughout the learning process. We demonstrate the effectiveness of our method in two areas: (i) stomatal detection and (ii) detection of costal and bulliform zones. Through extensive quantitative and qualitative experimental results on three crop species: *Setaria viridis*, *Sorghum bicolor*, and *Zea mays*, we show that our approach outperforms state-of-the-art segmentation methods in terms of both precision and time efficiency.

## 1 Introduction

Stomata are structures found in the plant epidermis, consisting of one or more cell types that control the opening of a microscopic pore. Through these pores, plants can assimilate carbon dioxide from the atmosphere and release water vapor via transpiration Willmer & Fricker (1996). In grasses, the stomatal complex includes the guard cells and two subsidiary cells to form an oval or diamond shape. Biologically, stomata play a crucial role in several vital plant processes, including but not limited to water use efficiency Leakey et al. (2019), pathogen entry Melotto et al. (2008), uptake of air pollutants Ainsworth et al. (2008), and temperature regulation Chaves et al. (2003). Conse-

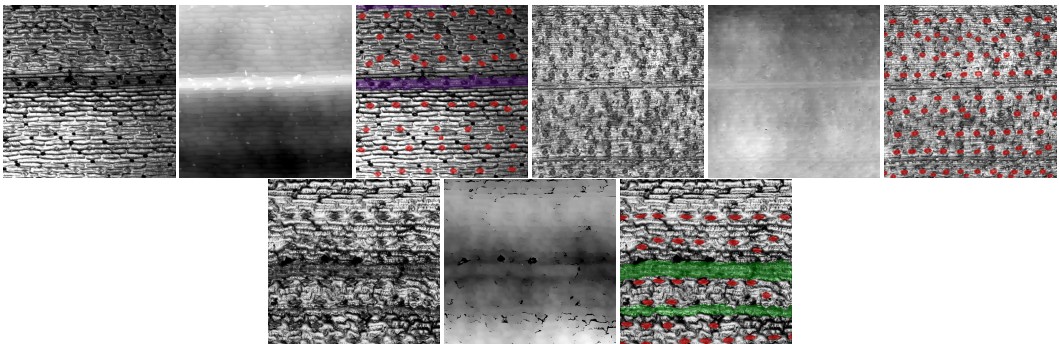

Figure 1: Visualization of the intensity, topography and the annotated ground truth for setaria abaxial (top-left three columns), sorghum abaxial (top-right three columns), and maize adaxial crops (bottom-row). The class legends for the ground truth annotation are as follows: red for stomata, violet for costal zones, and green for bulliform cells.

quently, stomata also affect the carbon and water dynamics of broader ecosystems Hetherington & Woodward (2003).

Quantifying stomatal traits from microscopic images is traditionally based on manual measurements, which require biologists with expert knowledge to accurately identify and assess stomatal morphology. In microscopic leaf images, the stomata are visually distinct from surrounding cells and generally exhibit homogeneous morphology within functional plant groups Serna (2011). This, combined with their biological significance and the time-consuming nature of manual data collection, has made stomata an increasingly challenging target for automated image analysis tools Tan et al. (2024). However, many of these tools exhibit a common issue: models trained on a particular species often perform poorly when applied to others Li et al. (2023); Jayakody et al. (2021); Song et al. (2020); Meeus et al. (2020); Casado-García et al. (2020). Even within a single species, variations in growth conditions, sampling methods, or imaging techniques can reduce model performance Li et al. (2022); Sakoda et al. (2019). Moreover, while the stomata of grass species typically appear in relatively regular rows parallel to prominent leaf veins, to our knowledge, no tools currently incorporate biological domain knowledge into the process to enhance accuracy Tan et al. (2024).

The Poaceae family includes all grass species, including economically and ecologically important crops such as maize, wheat, and sugarcane Barker et al. (2001). This paper discusses the application of an adaptable machine learning model to three species within this family: *Zea mays* (maize), *Setaria viridis, and Sorghum bicolor* (Fig. 1 shows the grayscale intensity and topography channels from an optical tomography microscope for each species). The appearance of a stomatal complex in mature tissue results from complex developmental pathways that are intrinsically connected to the development of other epidermal structures. Phenotyping these structures, in addition to the stomata, provides a much richer understanding of epidermal organization. A study by Ellis (1979) Ellis (1979) describes the terminology used in this paper to categorize the array of cell types and anatomical features observed in the leaf epidermis of the Poaceae.

Ellis identifies a large group of non-stomatal epidermal cells, sometimes referred to as pavement cells, and classifies them into two categories with distinct morphologies: intercostal (IC) cells and

costal cells. Costal zones are regions made up of costal cells and other cell types that occur in patterns consistent with the underlying vasculature of the leaf and are notably devoid of intercostal cells and stomatal complexes. The number and position of costal zones and have been shown to be correlated with the number of higher-order veins Aliscioni et al. (2016); Cerioli et al. (1994); Freeling & Lane (1994); Maricle et al. (2009). For this reason, phenotyping the costal zones as a whole, rather than segmenting the individual cells that make up them, is an excellent candidate for computer vision characterization to facilitate biological insight into vein density as a key hydraulic trait of plants Sack & Scoffoni (2013); Ueno et al. (2006). Bulliform zones are regions of bulliform cells, which Ellis describes as challenging to distinguish from intercostal cells by morphology. (Fig. 1 leftmost column shows intensity images of setaria, sorghum, and maize crops) and, like costal zones, completely lack stomata. Bulliform cells facilitate leaf blade rolling under water stress, with the number and size of bulliform zones correlated to the precise dynamics of leaf rolling Matschi et al. (2020). Rapid phenotyping of bulliform zones, in addition to stomatal traits, helps develop a more comprehensive understanding of plant water relations.

Numerous efforts have been made to automate the detection of cells and patches of cells. These efforts have shown that obtaining a general solution to the problem involves several challenges. The major challenges and how we address them in our method, presented here, are summarized below.

*Challenge 1: Lack of intrinsic model and unclear visual boundary-* The guard and subsidiary cells that make up the stomatal complex of the crops studied here often lack a more visually clear boundary. There is subjective variation in the boundary annotation of each cell, as there is no consistent correlation with color, intensity, or edges. As a result, individual cell instances are difficult to separate from the background. Delineating each cell instance becomes difficult because it often requires visually interpolating cell boundaries subjectively. Two experienced researchers could reasonably place the boundaries of a single cell in different places.

Likewise, cell categories such as costal and intercostal cells are primarily distinguishable by their geometry and position relative to the other cells within a leaf. However, when transitioning between costal and intercostal cell files, we often find cells that look like a hybrid of both types. Thus, there is subjectivity in cell shapes as well. Because the leaves are imaged from above, the apex of cells often reflects light into the objective and produces a highlight in the image. However, the curved edges of cells can be challenging to capture, resulting in a dark pixel boundary. In addition, the proximity of epidermal cells can further obscure the exact demarcation of the border of a subsidiary or guard cell. Debris and uneven highlights due to the microtopography of the leaf surface also lead to visually unclear and challenging boundaries between various cell instances.

*Approach:* We use multiple data sources, such as intensity and topography, to develop an inherent, intrinsic model that addresses the issue of unclear visual boundaries. Although topographic data alone are often subject to variations due to macrotextures, combining them with intensity data provides additional microtexture information that intensity data alone lacks. We begin by analyzing pixel-based evidence, which we then refine to obtain region-based evidence by considering the responses from neighboring pixels. This approach helps us identify all potential candidates for a target cell type, though it may include false positives at this stage. Additionally, we incorporate domain knowledge specific to each crop type and apply geometrical shape and size considerations to determine the best-fit boundaries for the target cell type.

*Challenge 2: No apparent spatial arrangement relationship* - The inherent variability of biological data, including factors such as shape, size, location, texture, and illumination within individual cells,

is substantial. For example, stomatal complexes are more elliptical in setaria but diamond-shaped in sorghum. Therefore, an intrinsic model alone is insufficient. We also need information about the spatial arrangement. However, the spatial relationships vary significantly between different cell types. While it is common to observe stomata in linear files parallel to the costal zones, deviations from this arrangement occur more frequently than expected. Furthermore, this spatial arrangement varies across species, meaning that it is not universally consistent.

*Approach:* To address this, we develop evidence based on the regions identified previously and incorporate global image-level considerations. At this level, we extract evidence from individual regions (individual cell instances) and analyze how they influence each cell's underlying geometric orientation and arrangement. For example, it has been observed that the probability of finding a costal zone near another increases as we move three times the distance of the width of the costal zone at hand. Similarly, the probability of finding stomata in a linear file is higher than finding one outside that file. Lastly, we also incorporate domain knowledge specific to the crop type. For example, in setaria leaves, no stomata are found within a costal zone. This eliminates all false positive candidates that have a high likelihood of being mistaken for stomata detected in the previous phase (this stage helps us eliminate most of the false positive candidates that were captured).

*Challenge 3: Texture-cluttered topographic data* - The topography data consist of both macro-textures and micro-textures that work together. The presence of macro-textures results in multiple high- and low-lying regions, which could arise from factors such as a folded or crinkled leaf, an uneven scoping surface, bends, or dents on the surface, etc. Microtextures provide the most important information and can highlight differences between embossed or etched cells. For example, stomata are low-lying cells, while hair and costal zone cells are outward-protruding regions. Hence, separating the macro-textures from the micro-textures is considered challenging. Fig. 2 shows the visualization of the macro and micro-textures along the epidermis (z-axis) of a leaf of setaria along the adaxial cross-section.

*Aproach:* The approach mentioned for handling challenge 1 takes care of this as well.

*Challenge 4: Not ML friendly* - In addition to the two significant challenges mentioned above, there exists a considerable amount of phenotypic diversity within these epidermal datasets from genetically diverse plant populations. Even for a single species, substantial manually annotated training data is required to achieve decent performance. Furthermore, these models often fail when applied to previously unseen species. As a result, significant labeled training data is needed to design an adequately performing ML-based model. Because of this, most existing models are crop-specific and cannot be generalized across different crops.

*Approach:* Due to the complex nature of plant biological data, creating a generalizable segmentation model for all genotypic and phenotypic plant types within a crop requires a vast amount of annotated training data. To address this, we propose *active annotation*. Active learning is an iterative process in which a machine learning model is trained initially with a small set of annotated data. The model is then used for inference on the unlabeled dataset. The inferred images are evaluated based on their performance, and the best-performing set of images are selected for the next round of classifier training. These images are added to the training pool, and the process is repeated iteratively until the desired model performance is achieved.

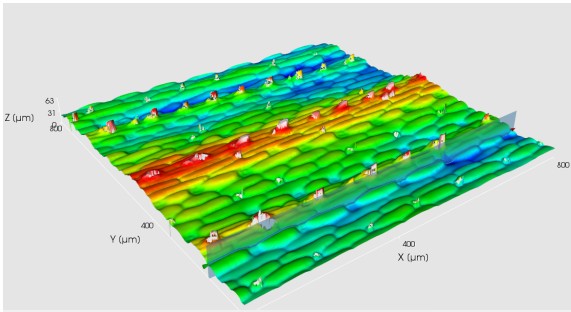

Figure 2: 3D Visualization of the macro and the micro textures along the epidermis (z-axis) of a setaria leaf along an adaxial cross-section. Its corresponding intensity and topography images are shown in the first row of Fig. 1. The hair cells and prickles are outward protruding regions (micro-textures), in addition to the overall high points of the leaf (macro-textures) resulting from an un-flattened leaf.

In this context, to assist the biological community in conducting stomata studies, we propose an automated strategy for classifying microscope images by species, detecting stomata, and segmenting other structures such as costal zones and bulliform cell zones. Our work is seminal because it significantly reduces the time required to examine stomatal traits, and it incorporates knowledge about the repetitive patterns of grass stomata and zones devoid of stomata to make human-like predictions. This study analyzes microscope images of setaria, sorghum, and maize. Some of our key contributions to this work are listed below.

1. Developing a cell model using multimodal data, where the primary modalities include RGB and depth (topography) images. Within the RGB modality, we use a specialized feature detector (a detector that identifies features applicable to any cell across all species). The intrinsic model learns from the unique points, shape features, topography map, and RGB data.

2. Separating the knowledge of spatial structure from the intrinsic cell models, which helps us reduce noise in the spatial distribution and structure of the cells. The spatial knowledge comes from the distribution of various bands across species, their linear arrangements, the regularity of the arrangement along the lines, the regularity of inter-band separation, etc.

3. Injecting domain knowledge for each crop type and enforcing geometric shape/size considerations to obtain the best-fit boundaries for the target cell type. Domain knowledge primarily includes the placement of different kinds of cells, bands, etc.

4. Compartmentalizing the domain knowledge from the generic model, allowing the system to be used for different species. To add new crop species, the domain knowledge block must be updated with crop-specific rules, while the remaining generic model stays the same.

5. Separating confusing classes (hair cells and stomata) using a separate classifier and active annotation, and reinforcing the training dataset with additional samples.

6. Designing a built-in system with a simple user interface that anyone can use. To add a new crop, only the domain parameters need to be entered, while learning and estimating all other parameters is part of the system.

## 2  Related work

This section discusses some of the existing pioneering works in this domain and identifies the research gaps. This part is divided into two main subsections: i) detection of stomata cells and ii) segmenting costal and bulliform zones. Stomatal conductance ($g_s$) put simply, represents the ease with which gases diffuse through the stomatal pore. Stomatal conductance is the product of a variety of epidermal traits including stomatal density, stomatal size, stomatal aperture (degree of opening of the stomatal pore), and stomatal index (the proportion of stomata relative to the total number of cells). Accelerating the phenotyping of any of the component epidermal traits will therefore benefit research into plant-environment interactions at the stomatal pore. We also highlight how our work differs from the rest and how we handle some challenges mentioned in the previous section.

**Stomata detection.** Images of maize, setaria, and sorghum utilized in this study were acquired as detailed in Xie et al. (2021), Prakash et al. (2021), and Ferguson et al. (2021), respectively. Previous work using this imaging method listed above has used intensity or a filtered version of the topography layer that smooths microtopography (detailed in Xie et al., 2021 Xie et al. (2021)) to bring the contours of cells into higher contrast. Unlike these analyses, this article uses both the images output by the microscope software, the intensity, and the topography layers (top row of Fig. 1 shows the intensity, topography and the ground truth images of a setaria leaf). The intensity layer is the product of surface light reflectance in the objective, displaying the visual appearance of the leaf epidermis. The topography displays the height of the z-axis of each pixel using pseudo-color. Each image layer provides different information that can be leveraged by the model, depending on whether a structure is distinct from its surroundings by appearance or height.

Tan et al., 2024 Tan et al. (2024) describe 39 articles using machine learning to phenotype stomatal traits, and since then, at least one more has been published Ku et al. (2024). However, the 40 models are supervised and require manual annotation to generate training data and manual validation to determine performance on test data. Furthermore, while extensive manual annotated training data can improve performance on the species depicted in those data, it can limit the model's performance on novel data. Stomata exhibit great phenotypic diversity in all taxa. However, consistent patterning and size make stomata excellent candidates for unsupervised learning techniques within one or a few related species.

**Costal zone and bulliform area segmentation.** The epidermal cells of grass leaves are often organized into two visually distinct zones running in longitudinal bands from the base to the tip of the leaf blade, termed costal zones and intercostal zones Ellis (1979). The costal zones are devoid of stomata and on the adaxial surface of maize leaves, lie over the major (lateral) veins, which allows the calculation of the density of the lateral veins from epidermal imaging Freeling & Lane (1994); Hay et al. (2000). Whether this pattern of direct correspondence between costal zones and lateral veins is consistent in all other grass species is unknown, however Maricle et al. Maricle et al. (2009) describes the presence of raised adaxial "ridges" over all major veins in 13 species of the $C_4$ grass genus *Spartina* and Aliscioni et al. Aliscioni et al. (2016), describes the presence of raised adaxial

"ribs" over first and second order veins in at least five species of the $C_4$ grass genus *Setaria*. The number of costal zones and veins seems to be correlated but species-specific.

Bulliform cells (BCs) are thin-walled cells with enlarged vacuoles that have been extensively studied in relation to leaf rolling as a response to water stress Hay et al. (2000); Moulia (2000). BCs in maize and rice occur in longitudinal zones, multiple cells wide on the adaxial leaf surface of adult-form leaves, and cross-sectional imaging suggests that they form over the space between intermediate veins Freeling & Lane (1994); Sylvester et al. (2001); Xiang et al. (2012); Matschi et al. (2020). Rice BCs have been associated with semi-rolled phenotypes in mutants where their size is reduced Hu et al. (2010), with abaxial rolling in mutants with increased bulliform cell number and size increased Zou et al. (2011), and with increased degrees of leaf rolling when their number is increased Xiang et al. (2012). In maize, abaxial rolling associated with an increase in the number and size of BC has also been observed Gao et al. (2019). In most papers characterizing bulliform cell traits, be it number, size, or shape, data are collected by manual analysis.

To our knowledge, only one paper has been published that shows the viability of AI-based analysis for either costal zone or bulliform cell phenotyping. It identifies the number and width of bulliform cell "columns" Qiao et al. (2019). However, similar to the previously described stomatal detection models, this model results from manual annotation of bulliform columns on 120 image training images. With regard to its wider application, it is subject to the same limitations as stomatal models. The fact that both costal zones and bulliform cells occur in these regular striped patterns lends them perfectly to this proposed unsupervised learning method for rapid phenotyping.

## 3 Method/Methodology

### 3.1 Problem statement

In this paper, we address two significant tasks that have remained challenging problems in plant biology:

**Detection of zones (costal/bulliform) from the microscopic images of a leaf epidermis:** The costal zones are in the vicinity of veins, which is valuable to understand plant water relations. Leaf vein density is an important trait for understanding a plant's hydraulic capacity and its tolerance to stresses on the vascular system Sack & Scoffoni (2013). Because of its correlation with different orders of leaf veins, phenotyping costal zone density and size across the leaf surface may provide a quick and simple proxy measurement for leaf vein density. The role of bulliform cells in a plant's water stress response, as described above, suggests that this trait could benefit from rapid, unsupervised data collection to replace costly manual measurements. Due to the similar visual appearance of costal and bulliform cell zones in grass leaves as distinct longitudinal sections, a single tool can be applied to phenotype both traits.

We have designed a costal zone detector for different types of crops. Using a limited number of leaf patches, we trained a classifier to predict the probability that a costal zone occurs at a specific location in image. By combining this with domain knowledge of leaf type, we incorporate geometric constraints and spatial structure information into the candidate costal zones to refine the results.

**To detect stomata cells from microscopic images of a leaf epidermis:** Plants with higher stomatal density grow faster under optimal growth conditions. We have designed a model to detect

all instances of stomata within a leaf image. While the intensity image of the leaf captures potential stomatal candidates well, there is significant confusion with hair cells. In the topography data, however, these two classes are distinguishable. We use this condition, along with geometric constraints and spatial structure information, to select the stomata. Finally, we combine the evidence from both the topography and intensity images to obtain the final results.

We have designed a robust system to automatically detect costal zones and stomata in any given grass crop using minimal training data.

## 3.2 Preliminaries

We propose a generalized framework to handle different species of grass crops using limited annotated training data, depending on the task at hand. The proposed framework consists of four main submodules: i) active feature learning block, ii) geometric consideration block, iii) evidence maximization block, and iv) domain knowledge injection block. The active feature learning block, the geometric consideration block, and the evidence maximization block are general modules that remain fixed across all types of crops. Depending on the selected crop, only the domain knowledge injection block has hyperparameters that change, while the overall steps remain the same. The submodules also remain consistent for different tasks (e.g., costal zone segmentation, bulliform zone segmentation, or stomata detection). In the following section, we elaborate on each of these modules.

### 3.2.1 Active feature learning (gray block)

This stage involves learning evidence at the pixel level. The learning block uses a limited set of annotated training data to train an initial classifier. The classifier is trained to use features computed by applying local filters and estimates a pixel's probability of belonging to one of the cell types. Upon model deployment in the testing phase, we import the crop-specific classifier and use it for inference. This classifier is updated using *active learning*. Active learning is an iterative process of training a machine learning model, starting with a small set of annotated training data. The model is then used for inference on the unlabeled data set. The inferred images are evaluated based on their performance. The best-performing set of images is selected for the next round of classifier training, and these images are added to the pool of training data. The active learning block helps reinforce the training data with additional annotated samples, thereby improving the overall performance of the framework. This approach addresses the model's dependency on large training data and aids in categorizing hybrid cells. This block helps solve challenges 1, 2, and 4 from Section 1.

### 3.2.2 Geometrical Statistics (blue block)

This stage involves learning evidence at the regional level. In the previous active feature learning block, we treated each pixel as an individual point, which was assigned a probabilistic value for being classified into different cell types. At this stage, we do not consider neighborhood information. The probability of finding another pixel of the same cell type near a given pixel is much higher in natural image statistics. In the geometric statistics block, we begin treating each cell as a region and merge all individually handled pixels into groups, which serve as the cores of the different cell types. This process greatly helps refine the initial predictions by incorporating neighborhood properties. This block primarily assists in cleaning the regions, suppressing tiny components, separating individual

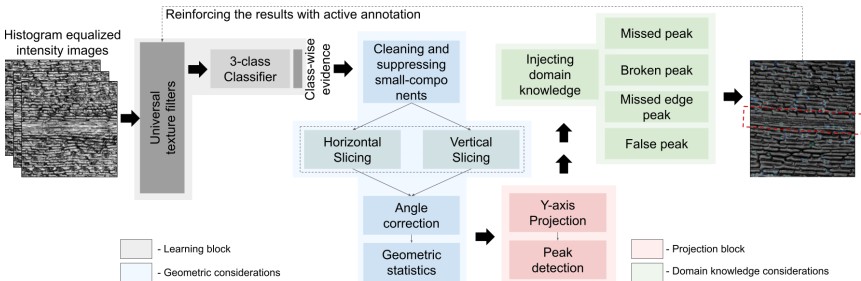

Figure 3: Figure illustrates the overall pipeline of the proposed costal zone detector. The images are passed through universal texture filters, and their responses are used to train a pixel-wise classifier (gray block). The evidence from this stage is then used to refine the results in the subsequent blocks, where: i) geometric considerations are made (blue block), ii) cumulative projection along the vertical axis detects the region with the maximum evidence of the costal zone (pink block), and iii) domain knowledge is injected (green block).

cell instances from the overall semantic segmentation, and applying other geometric structural constraints. This block helps address challenges 1 and 2 from Section 1.

### 3.2.3 Evidence Maximization (pink block)

The previous block helps transition from individual pixels to regions and provides evidence values for each cell type within those regions. In this block, we move from a regional level to a global image level, identifying the regions that provide the maximum evidence of occurrence. At this level, we reassign weights to the various evidence found across different regions and determine which ones to retain and which ones may be erroneous detections. This process helps eliminate a significant number of false positives and reconsider some of the missed true negative candidates. This block helps address challenge 3 from Section 1.

### 3.2.4 Incorporating Domain Knowledge (green block)

Finally, we examine the geometric and photometric considerations in this block. Depending on the task and crop, this block is updated with crop-specific and task-specific parameters. When handling a new crop or task, this is the block where crop/task-specific domain knowledge is required. Through hyperparameter tuning and statistical analysis of the task, this block can be customized.

In the following two sections, we will provide detailed explanations of each of these with respect to the problem statements mentioned above.

## 4 Problem statement 1: Detection of Costal Regions

### 4.1 Active feature learning block

**Input data selection and processing:** The microscope provides both intensity and topography data for leaf images. While the intensity image has its advantages, the topography data is also crucial for this task, as it helps differentiate between the two confusing classes: hair and stomata cells. Typically, hair cells are outward-protruding regions and are very distinctive in the topography

data. However, to avoid the bias of unnecessarily learning the macrotextures from the topography data, we train two separate models: one using intensity data and the other using topography data. We perform histogram equalization on the raw intensity data to make it more uniform.

**Creation of the ground-truth** The available ground truth annotations included IC cells, stomata cells, and costal zones. We used the provided annotations, but they had slightly overlapping boundaries with adjacent cells. To improve separability and learn the ridges and canyons more effectively, we required at least a 1-pixel separation between adjacent cell types.

1. Case I: When there is $\geq 1$ pixel separation: Do not do anything in this case.

2. Case II: When there is overlap between the annotated boundaries of adjacent IC cells: Shrink both boundaries inward in the direction of overlap until there is at least 1 px separation.

The resulting image serves as our ground truth, where each pixel belongs to either an IC cell or the background class.

**Costal and intercostal cell response**: From the histogram equalized topography data of the leaf images, we use a set of $(2k + 1) \times (2k + 1)$, filters (k = 1, 2, 3), to identify centers of uniquely textured neighborhoods, such as done by SIFT Lowe (2004) point detectors, corner detectors Harris et al. (1988), etc. So we used filters of sizes $3 \times 3$, $5 \times 5$, and $7 \times 7$, as the maximum height of the IC and stomata cells is $\approx 10$ to 13 pixels. We apply these filters to the histogram-equalized intensity image $I$ and obtain evidence from the classifier $\theta(\cdot)$ as $\theta(I)$.

We balance the $(+)^{ve}$ examples from the cells and the $(-)^{ve}$ examples from the background using class weights. This approach assigns higher weights to the boundary and $(+)^{ve}$ regions. We concatenate all the input channels and train a point-wise image classifier to distinguish between two classes: background and IC cell. Since IC cells make up the majority of the image, we train a binary classifier using an artificial neural network.

We learned a classifier with 85% accuracy in the validation images. We then choose all regions which are $\geq 75\%$ confident that the region is a coastal zone or a background region and get initial estimates of possible IC cores (if we want to segment all instances of IC cells, too).

## 4.2 Geometrical considerations block

This block helps to delineate the costal zones identified by the classifier based on their geometry. We use a series of steps to refine the initial estimates of the evidence obtained from the classifier's response, as described below.

1. **Cleaning:** We apply a Gaussian filter to smooth the response image and retain all connected components with an area $\geq N$ pixels. We refer to this operator as $P(\cdot,\text{N})$.

2. **Dis-joining connected components:** From the intercostal cell response, segmenting the individual instances of IC cells helps estimate the overall angle of orientation of the costal zones. To achieve this, we perform horizontal and vertical slicing to obtain rough initial estimates of the IC cells. (i) Vertical slicing: We compute the average length of the bounding boxes of the predicted cores. If any bounding box length > (avg + n), we divide

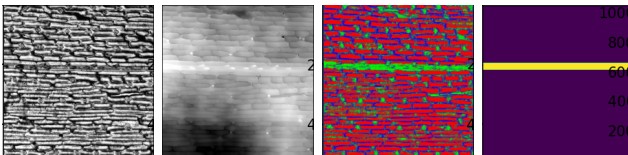

Figure 4: First and second images depict the intensity and topography images of a setaria leaf. The $3^{rd}$, and $4^{th}$ show the output of the active feature learning and the final zone prediction.

the bounding box into two parts from the center. (ii) Horizontal slicing: This step requires separating rows. For each row, we calculate the median and variance of the width and split the length of the bounding box at the thinnest region near the median. Together, the vertical and horizontal slicing operations are defined as $\phi(\cdot)$. The slicer function is applied to the pre-processed images $\theta(I)$ as $\phi(\theta(I))$.

3. **Angle correction:** Once we obtain the rough individual instance cores of the intercostal cells, we apply an ellipse fitting procedure to each of the connected components to determine the orientation of the major axis of all the IC cells. Without loss of generality, we assume that the IC cells are oriented roughly toward the costal zones, as they tend to align parallel to them. We compute the median angle of orientation for all the instances of IC cells and use this as the correction angle for the input images, ensuring that the costal zones align accordingly.

4. **Geometrical structure** Since we use a pixel-wise classification backbone for the segmentation problem, we obtain many pixels at random locations that provide strong evidence for potentially belonging to a costal zone area. However, when we examine their neighborhood and find insufficient evidence for them to belong to a costal zone, we consider these pixels as noisy detections. For a region to be considered a potential costal zone area, substantial evidence from neighboring regions is required to strengthen the decision. If the predicted regions are too small, it is more likely that they represent just noisy evidence. We perform connected component analysis and remove all regions with an area smaller than $\approx 70$ pixels.

### 4.3 Evidence maximization block

After gathering evidence for the costal zones, we make angle corrections on the predicted data to align the zones parallel to the horizontal axis. We then calculate the cumulative strength of the evidence by summing the confidence scores along the y-axis. Ideally, if the evidence calculation is accurate, we expect sharp peaks when plotting this cumulative evidence along each row. However, due to the lack of sufficient training samples, the predictions $\mathcal{R}_{IC}$ are not very accurate in the various crop phenotypes and genotypes. This results in many additional noisy peaks. Furthermore, maintaining a constant threshold value for peak detection can often lead to missed or false coastal zone detections. To minimize this, we apply a Savitzky-Golay filter Savitzky & Golay (1964) for mean filtering of this signal. The peak heights of the spectra are better preserved with Savitzky–Golay filtering compared to other filters with similar noise suppression. Once we have the filtered response from the y-axis summation of the evidence, we heuristically choose a threshold value and select all regions above the threshold as potential costal zone candidates.

### 4.4 Domain knowledge injection block

Once we have all the candidates for the costal zones, we observe that there could be four possible cases of false or missed detection of the costal zones. However, since we have domain knowledge about the underlying geometry of costal zone occurrences, we inject this information as domain knowledge to further refine the predictions. The following steps are taken:

1. **Missed detection:** It has been experimentally observed that the spacing between one costal zone and the next is approximately three times the width of the thicker costal zone. Therefore, within an image, once we detect a highly confident costal zone, we lower the threshold as we move vertically away from the detected zone. This increases the likelihood of finding other under-confident costal regions around that location. Fig. 5(a) shows this response.

2. **Broken detection:** It has often been observed that a sharp peak breaks into two. This is mainly due to the presence of a long horizontal striated region of silica cells, which gives a low response to the costal zones. As a result, the peak breaks at the center. To compensate for this, whenever we detect a broken peak (i.e., the occurrence of two sharp, highly confident peaks in close proximity, typically less than a few pixels apart), we retain only the most prominent peak and extend it across the costal zone. Fig. 5(b) shows this response.

3. **Missed edge zones:** The edge regions of the image tend to give a lower response to the costal zone detectors. When examining the edge of the image, provided that the distance to the nearest detected costal zone is approximately three times the width of that zone, we lower the threshold for peak detection. Refer to Fig. 5(c).

4. **False positive detections:** When there are too many peaks that exceed the threshold value, we select the most prominent peaks (the prominence of a peak measures how much it stands out due to its intrinsic height and location relative to other peaks). We retain only the most prominent peaks initially, then suppress the non-prominent peaks that are too close to a detected costal zone (where *closeness* is defined as a distance relative to the width of the costal zone). Refer to Fig. 5(d).

Taking into account the conditions mentioned above, we have developed a robust costal zone detection model that generalizes well across various genotypes and phenotypes of the crop species tested.

## 5 Problem statement 2: Stomata detection

### 5.1 Active feature learning block

**Input data selection and processing:** The microscope provides us with intensity and topography data of leaf images. Stomata are typically low-lying topographic cells in the leaf epidermis layer; however, the presence of macrotextures, along with microtextures, results in multiple high- and low-lying regions. Due to these macrotextures, some low-lying regions appear higher in the z-plane than the actual stomata. When we use raw (unflattened/unfiltered) topography data as

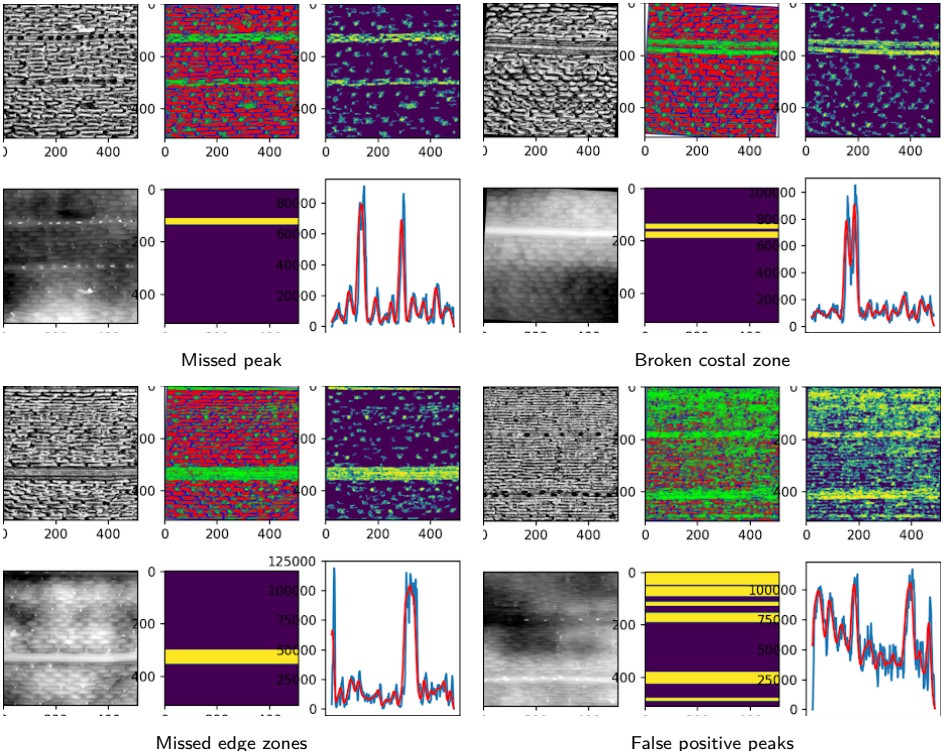

Figure 5: Different kinds of errors in costal zone detection that are solved by the domain knowledge injection module, shown for a setaria leaf. Left coloumn shows the intensity and topography images of a setaria leaf. The $2^{nd}$, $3^{rd}$ and $6^{th}$ shows the output of the active feature learning, geometric considerations, and evidence maximization block, respectively. The $5^{th}$ plot shows the final prediction from the domain knowledge block.

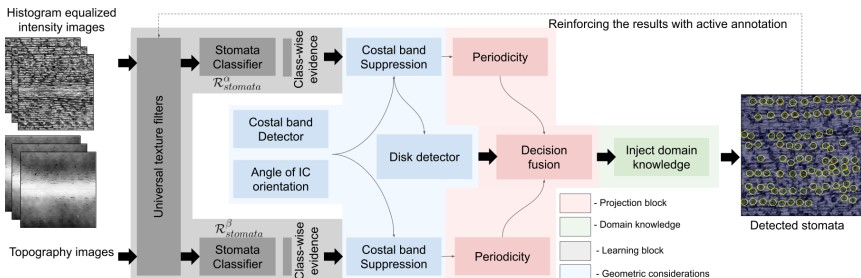

Figure 6: Figure illustrates the overall pipeline of the proposed stomata detector. The images are passed through the universal texture filters, and their response is used to train a point-wise classifier (gray block). The evidence from this stage is then used for refining the results in the subsequent blocks, where i) geometric considerations are made (blue block), ii) cumulative projection along the vertical axis to detect the region of maximum evidence of costal zone (pink block), and iii) injecting the domain knowledge (green block).

input and attempt to learn the stomata cells, the classifier tends to place significant weight on all the low-lying points, classifying them as potential stomata. However, the topography data helps differentiate between stomata and hair cells, as both occur embedded within intercostal cell regions. Hair cells are outward-protruding structures, while stomata cells are embedded deep within the folds of intercostal cells. This topographic difference distinguishes these two visually similar classes from the intensity image.

We train two separate pixel classifier models for intensity and topography data to avoid the bias introduced by macrotextures in the topography. To make the intensity image more uniform, we preprocess the raw intensity data by performing histogram equalization.

**Training a classifier:** We apply the unique neighborhood identification filters to the histogram-equalized topographic data from leaf images. Then, we apply these filters to the histogram-equalized intensity image $I_{intensity}$, and obtain the response $\alpha(I_{intensity})$. We trained a classifier that achieved $73.56\%$ accuracy on the validation images. Next, we select all regions with $\geq 75\%$ confidence that they are either stomata cells or background regions, and use this to generate initial estimates for potential stomata cell candidates. The response to the stomata cells is denoted as $\mathcal{R}^{\alpha}_{stomata}$.

We then apply it to the topography image $I_{topography}$ and get the response $\beta(I_{topography})$. We learned a classifier with $70.17\%$ accuracy on the validation images. We choose all regions that are $\geq 70\%$ confident and obtain initial estimates of possible candidates for stomata cells. The response to the stomata cells is denoted as $\mathcal{R}^{\beta}_{stomata}$.

### 5.2 Geometrical considerations block

The stomata cells are detected by combining the responses of i) the image and topography models separately, along with ii) the response of the disk detector.

**1. Disk detector response:** We use the multiscale region detector proposed in Blostein & Ahuja (1989). It is designed as a coarse estimator of the size and shape of regions. The detector identifies all image regions with minimal gray-level variations relative to its neighborhood. It has been found to be better suited for texture element extraction than for general image segmentation. The detector works by identifying circular disks that best fit uniform image regions. The diameter (D) is given by:

$$D = 2\sigma\sqrt{\sigma(\frac{\partial\{\nabla^2 G * I\}}{\partial\sigma})/(\nabla^2 G * I) + 2} \tag{1}$$

where $\nabla^2 G$ denotes Laplacian of Gaussian of image $I$, $\sigma$ is the value used for calculating the difference of Gaussian.

**2. Response of $\mathcal{R}^{\alpha}_{stomata}$ and $\mathcal{R}^{\beta}_{stomata}$** (injection of domain knowledge): We have domain knowledge about the intrinsic geometric structure of a leaf, which is biologically known to us. This knowledge can be incorporated and injected into the predicted model as a post-processing step.

1. We apply a Gaussian filter to smooth the response image and retain all connected components with an area of $\geq N$ pixels. We define this operator as $P(\cdot, N)$.

2. Since we are using a pixel-wise classification backbone for a segmentation problem, we obtain many pixels at random locations, providing strong evidence for belonging to a potential costal zone area. However, when we examine its neighborhood and do not find sufficient

evidence for it belonging to a costal zone, we consider this pixel a noisy detection. For a region to be considered a potential costal zone, we require considerable evidence from the surrounding regions to strengthen the decision. If these predicted regions are too small, it is more likely they are just noisy evidence. We perform a connected component analysis and remove all regions $<\approx 50$ pixels in area.

3. Costal zone suppression: We know that stomata cannot occur within a costal zone. Therefore, even though there is significant confusion between stomata cells and prickles in the intensity image domain for some crops like setaria, they can be filtered out by creating a hole-y image that excludes the costal region.

4. Finally, we perform a cleaning operation for the stomatal candidates based on their geometry. We fit ellipses to each of their contour, and if their major and minor axes are $< (T_1, T_2)$, we discard them. The thresholds $(T_1, T_2)$ are chosen by studying the individual crops.

### 5.3 Evidence maximization block

Once we have evidence from three domains—i) topography data, ii) intensity data, and iii) disk detector—some of the detections may be false positives, while others may be true negatives that have been missed. We use these predictions to make a combined decision.

**Decision fusion:** In each domain, we assign a high probability of finding stomata at the center of the predicted contour under consideration and gradually decrease the probability values as we move away from the center. This is similar to creating heatmaps for potential stomatal candidates. We then combine the heatmaps generated from the three domains. All detections that align in the union of the three domains are considered stomata cells. The remaining predictions that are i) geometrically small, ii) non-overlapping in the other two domains, and iii) have low-probability predictions are most likely false positives and are not considered stomata.

**Periodicity:** While the process described above helps eliminate a majority of false positives, we may still miss some true negatives. One important property that comes into play here is the periodic nature of stomata distribution. To address this, we use the sliced IC cells from the geometric considerations block of costal zone detection. The width of the IC cells in each file provides a rough estimate of the interval at which stomata are expected to appear.

### 5.4 Domain knowledge injection block

In this block, we incorporate all the crop-specific parameters, some of which are detailed under model architecture (Section 6.4). Additionally, we observe that stomata often get confused with hair cells. Hair cells are frequently misidentified as stomata cells. Hair cells occur outside the costal zones and share similar texture, shape, and size with stomata. The key distinction between them lies in their topography: hair cells are outward-protruding, whereas stomata are embedded within the ridges of two pavement cells. Both occur in linear files, but at different rows. While the topography data largely help reduce this confusion, the presence of macro-textures within the topography data can still result in some ambiguous cases. We address these issues in the following two ways:

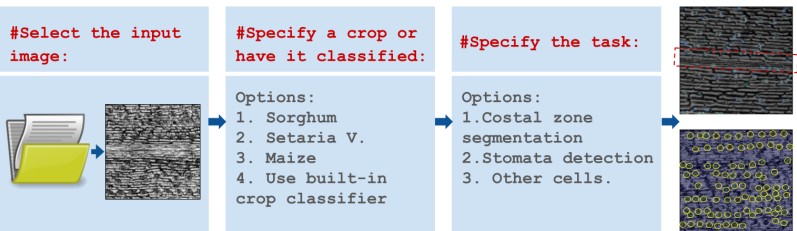

Figure 7: The overall pipeline of the proposed system and its corresponding user-interface for non-experts of plant biology.

1. All stomata are observed to occur in near-linear files. We leverage this domain knowledge by removing all predictions that are not linearly aligned (outside a neighborhood deviation of $\epsilon$) and have a low confidence score.

2. To minimize erroneous detections of hair cells being falsely labeled as stomata, we employ active feedback control to address these challenging cases and improve the next training batch.

## 6  Implementation details

### 6.1  GUI Interface

We have developed a system that allows any non-expert who does not have any background in plant biology, to phenotype stomatal density and costal zones. Figure 7 illustrates the system's pipeline. The user begins by uploading an image from the database. Once the image is selected, the user can specify the crop type, provided they have the necessary expertise. If not, the system offers an in-built classifier to automatically identify the crop type (details on this process are provided in the following section). Once the plant is categorized, the system loads the appropriate crop-specific parameters and model weights into the backend. The user is then prompted to choose the task to perform: i) stomata detection, ii) zone detection, or iii) detection of other cell types. For the first two options, the corresponding model (as described in Section 4) is loaded. For the third option, the user can perform active annotation on a few images and input crop-specific parameters. Some of the additional, extendable cell classes include prickles, pavement cells, macro-hair, silica cells, etc. The system then detects all the specified cell types throughout the image, providing the user with insights into the overall phenotype.

Crop classifier: We use our unique neighborhood filters of size $(2k+1) \times (2k+1)$ (k = 1, 2, 3, 4) to the intensity images and concatenate their responses. The concatenated responses are then fed to a simple convolutional neural network(CNN), which is optimized using a categorical cross-entropy loss function and a stochastic gradient descent optimizer. The model outputs the most probable crop class. Fig. 8 illustrates the overall pipeline of the crop classifier. The classifier has been trained on $\approx 10$ sample images of each crop, achieving an accuracy of 97.06% on the test data.

### 6.2  Image acquisition

Images of maize, setaria, and sorghum utilized in this study were acquired as detailed in Xie et al. (2021), Prakash et al. (2021), and Ferguson et al. (2021), respectively. Excised sections

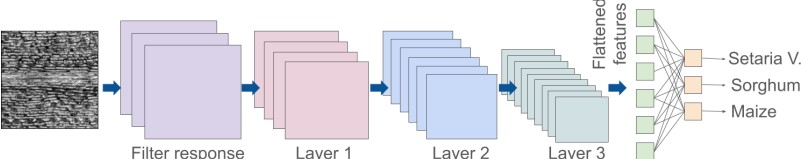

Figure 8: Convolutional neural network based model for crop classification using the response of texture filters response of the intensity images.

of fully expanded adult leaves were secured onto microscope slides with double-sided tape and imaged using the Nanofocus $\mu$surf Explorer Optical Topometer (Oberhausen, Germany) at 20X magnification. Each image has a resolution of $512 \times 512$ pixels, representing an area of 0.64 mm$^2$. Previous studies using this imaging method, as referenced above, employed intensity or a filtered version of the topography layer that smooths the microtopography (as detailed in Xie et al., 2021 Xie et al. (2021)), enhancing the visibility of cell contours. For our analysis, we use both the intensity and topography layers. To assess the model's performance, expert biologists manually annotated stomata, intercostal cells, costal zones, and bulliform cell zones using the VGG Image Annotator Dutta & Zisserman (2019), as detailed in Xie et al., 2021 Xie et al. (2021).

### 6.3 Datasets

**Stomata detection:** We evaluated the performance of the proposed framework on the setaria, sorghum, and maize datasets. For each leaf sample, both intensity and raw topography images were used. In setaria, there were 9 annotated stomata images, and for the test set, we used 1,714 images for stomata detection. In sorghum, 21 annotated stomata images were available, with 19,217 test images for detection. For maize, we used 20 annotated stomata images, with 640 test images for stomata detection.

**Zone segmentation:** For detecting important zones within the leaf, we used 12 annotated costal zone images of setaria and 1,714 unannotated test images (the same as the stomata test set). In sorghum, zones are not prominent. Hence, we did not perform zone detection for sorghum. For maize, we used 20 annotated bulliform zone images. The same 640 test images used for stomata detection were also used for bulliform zone segmentation.

**Crop-classifier:** For the crop classifier, we randomly selected 20 images from each of the three crop types and validated the model using the remaining images from the three datasets. The setaria images were taken from a random selection of a RIL population described in Prakash *et al.*, 2021 Prakash et al. (2021), the sorghum images were taken from a selection of accessions described in Ferguson et al., 2019 Ferguson et al. (2021), and the maize images were taken from a RIL population described in Xie et al., 2021 Xie et al. (2021).

### 6.4 Model architecture

As illustrated in Fig. 7, the user is first prompted to choose an image from the system UI. Once the leaf image is selected, the model uses the crop classifier to automatically identify the crop type. Afterward, the user can specify the task they wish to perform, such as: i) Costal zone segmentation, ii) Bulliform zone segmentation, or iii) Stomata detection. Based on the selected

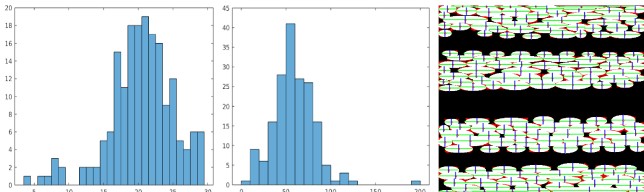

Figure 9: Plot of the number of IC cells along the y-axis vs the height and width of the individual IC cells for a setaria leaf in left and middle plots. Right plot shows the maximally fitted ellipses (white) along the IC cells (red) and the major and minor axes are represented in green, and blue.

task, the crop-specific model for that task is activated in the back-end. These tasks can also be combined in sequence. Each crop task module consists of two main components: i) The first part is the generic framework, which is common across all crops. ii) The second part includes domain-specific considerations, tailored for the chosen crop. The texture-based evidence predictor (gray) and geometric statistics (blue) are consistent across all crops and tasks. On the other hand, the evidence maximizer (red) and domain knowledge injector (green) have the same backbone architecture, but differ in crop-specific hyperparameters. These hyperparameters are invoked when the user selects the crop and task. In the following subsections, we will detail the hyperparameters specific to each crop.

**Setaria:** exhibits well-defined costal zones, as shown in Fig. 9, which illustrates the average height and width dimensions of each pavement cell in this crop type. Individual IC cells are marked in red and overlapped with a tight-fitting ellipse (shown in white). This IC cell analysis is crucial for identifying IC cell files and performing row and column separations, which are then used to estimate the angle of orientation of the costal zones. Additionally, it has been experimentally observed that in setaria, the likelihood of finding a costal zone in the vicinity of another increases as we move three times the width of the current costal zone. Similarly, the probability of finding a stomata in a linear file of cells is higher compared to finding it outside that file. We set a threshold value of at least 7,000 for potential peak detection after Savitzky-Golay filtering. ig. 10 illustrates the intermediate steps involved in detecting the costal zones.

For stomata detection, statistical analysis shows that the maximum width along the major axis is ≈45 pixels (along the $x$ axis), and the minor axis is ≈30 pixels (along the $y$ axis). Furthermore, the difference between the major and minor axes ≤ 15 pixels. These values are used to clean false positives by applying geometric considerations based on domain knowledge.

**Sorghum:** Sorghum does not have prominent raised costal zones on the abaxial surface. However, we do observe bands where there are no stomata cells. We heuristically choose a very high threshold value (200,000) for peak detection to avoid wrongly detecting these bands. In absence of prominent costal regions, cell detection is more straightforward.

In sorghum, stomata cells exhibit a diamond-shaped structure. Through statistical analysis, we find that the maximum width along the major axis is ≈50 pixels (along the $x$ axis), and the minor axis measures about 40 pixels (along the $y$ axis). The difference between the major and minor axes ≤15 pixels. These dimensions are used to clean up false positives through geometric considerations based on domain knowledge.

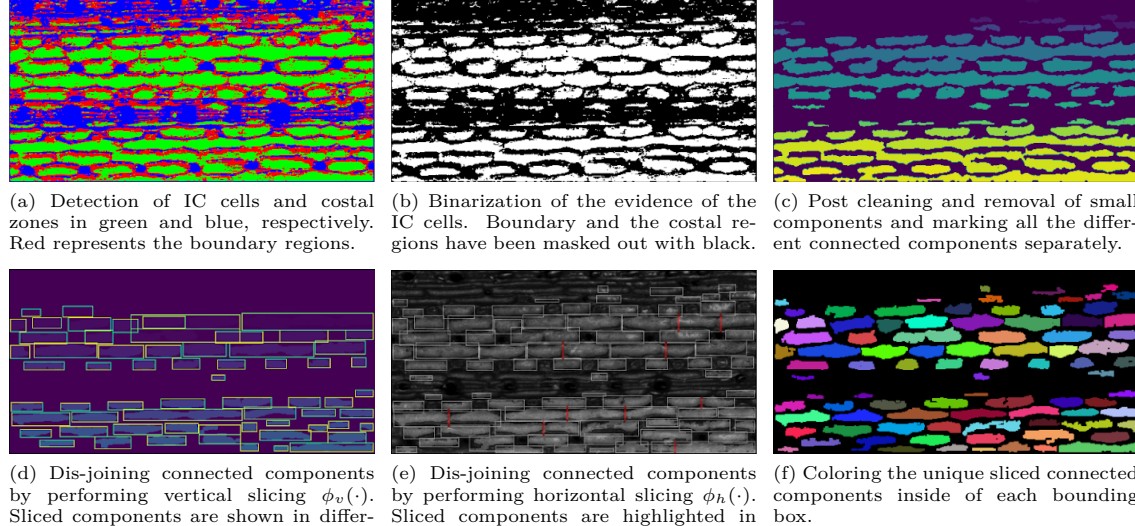

(a) Detection of IC cells and costal zones in green and blue, respectively. Red represents the boundary regions.

(b) Binarization of the evidence of the IC cells. Boundary and the costal regions have been masked out with black.

(c) Post cleaning and removal of small components and marking all the different connected components separately.

(d) Dis-joining connected components by performing vertical slicing $\phi_v(\cdot)$. Sliced components are shown in different boxes.

(e) Dis-joining connected components by performing horizontal slicing $\phi_h(\cdot)$. Sliced components are highlighted in red.

(f) Coloring the unique sliced connected components inside of each bounding box.

Figure 10: Visualization of some of the intermediate steps within the learning block and the geometric statistics consideration box from Fig. 3 for costal zone detection on a setaria leaf.

**Maize:** Similar to setaria, maize also has well-defined, thick costal zones that are visually distinct. In maize, these zones are primarily formed by the Bulliform area. Since the Bulliform areas exhibit a sharp contrast in comparison to the surrounding regions, peak detection is relatively straightforward. A threshold as low as 2000 works effectively for this dataset.

For stomata detection in maize, the major axis is set to ≈45 pixels, while the minor axis is set to 30 pixels. The maximum allowable difference between the major and minor axes is 15 pixels. These values are used to help eliminate false positives by leveraging geometric considerations based on domain knowledge.

### 6.5 Training and evaluation protocol

While geometric and domain-specific considerations help address the significant diversity between genotypes and phenotypes within a crop species, the performance of the classifier remains constrained by the limited training data available. To overcome this limitation, we utilize active learning. We begin by training crop-specific models using the initial training dataset to create an initial classifier. This classifier is then integrated into the full model and used to infer results on the remaining test data. Next, we manually select 10 images that perform well in terms of Intersection over Union (IoU) scores and recall values. These informative images are then chosen for the next annotation round and added to the training dataset. This iterative process continues until the model achieves satisfactory performance across a broad range of crop images. We conducted three rounds of active annotation for setaria, two rounds for sorghum, and four for maize, achieving the desired performance.

For evaluation, we use precision, recall, and IoU to compare the detected stomata cells/zones against the ground truth. Additionally, for the remaining dataset, we use mean square error (MSE) and mean absolute error (MAE) to assess performance. MAE measures the average absolute difference between predicted and ground truth stomatal density, treating both missed and false detections equally and being less sensitive to outliers. MSE calculates the average squared difference, penalizing larger errors more heavily, and is reported as a percentage for better interpretability.

Table 1: Performance of the proposed model and SAM on the gold standard dataset for the detection of stomata using Precision, recall and IoU.

| | Precision | | Recall | | IoU | |
|---|---|---|---|---|---|---|
| Plant | SAM | Ours | SAM | Ours | SAM | Ours |
| Setaria | 0.708 | 0.918 | 0.659 | 0.879 | 0.261 | 0.849 |
| Sorghum | 0.683 | 0.844 | 0.706 | 0.893 | 0.435 | 0.782 |
| Maize | 0.753 | 0.851 | 0.743 | 0.902 | 0.483 | 0.622 |

Table 2: Performance of the proposed model and SAM on the gold standard dataset for the costal zone detection using recall and IoU.

| | Recall | | IoU | |
|---|---|---|---|---|
| Plant | SAM | Ours | SAM | Ours |
| Setaria | 0.092 | 0.951 | 0.049 | 0.931 |
| Maize | 0.102 | 0.996 | 0.114 | 0.883 |

## 7 Experiments

In this section, we present the overall experimental results achieved by the proposed algorithm and compare them with the state-of-the-art Segment-Anything Model (SAM) Kirillov et al. (2023), while also performing ablation studies.

### 7.1 Performance and comparison with existing algorithms

The SAM model is trained on a massive dataset consisting of 1 billion masks across 11 million images. This extensive database allows SAM to perform zero-shot segmentation on completely unseen data. In contrast, our model is trained with a limited amount of labeled training data (less than 15 images per leaf species). To ensure a fair comparison, we also apply the domain-knowledge-based consideration block to the output of SAM to refine its segmented results.

We evaluated both SAM and our proposed model for leaves from three species for two tasks: stomata detection and costal/bulliform zone detection. We conducted two sets of experiments. The first used gold standard data, which were manually annotated by biologists and are considered highly accurate. This dataset consists of 100 test images for sorghum. The second set involves large-scale data annotated by volunteers with no domain expertise, though they underwent a short training session. These annotations may not be 100% accurate. We report results from both datasets.

In the sorghum gold-standard dataset for stomata detection, SAM achieves an IoU of 0.435, while our proposed algorithm yields a significantly higher IoU score of 0.782. Moreover, our algorithm surpasses SAM by approximately 18-20% in both precision and recall. Qualitative and quantitative analysis reveals that while SAM effectively delineates the boundaries of detected stomata cells, achieving high IoU for correctly detected cells, it struggles with detection. SAM misses many stomata cells and produces a high number of false positives, leading to a lower overall IoU. For setaria, stomata detection proved more challenging for both models, but our model outperformed SAM across all crop types and evaluation metrics. In the larger dataset of all samples, we observed

Table 3: Performance of the proposed model and SAM on the full dataset for stomata detection and costal zone detection using MAE and MSE.

| | Stomata detection | | | | Costal zone detection | | | |
|---|---|---|---|---|---|---|---|---|
| | MAE | | MSE (%) | | MAE | | MSE (%) | |
| **Plant** | SAM | Ours | SAM | Ours | SAM | Ours | SAM | Ours |
| Setaria | 23.38 | 8.90 | 48.4 | 11.66 | 2.03 | 0.12 | 98.06 | 0.93 |
| Sorghum | 15.49 | 4.69 | 26.0 | 9.57 | - | - | - | - |
| Maize | 12.35 | 4.77 | 21.8 | 8.67 | 0.97 | 0.00 | 99.10 | 0.00 |

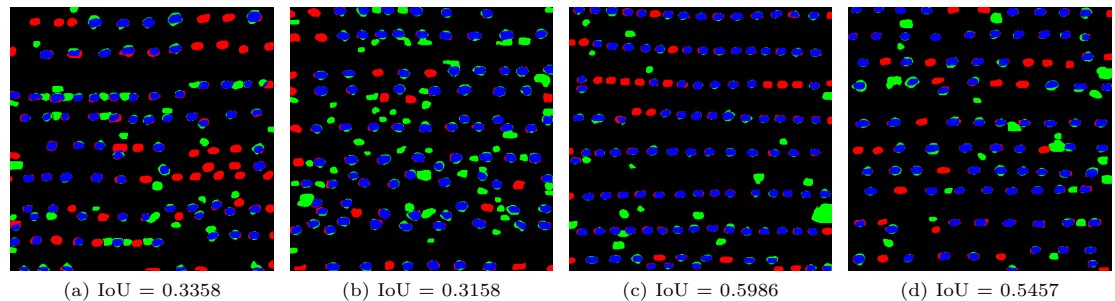

(a) IoU = 0.3358    (b) IoU = 0.3158    (c) IoU = 0.5986    (d) IoU = 0.5457

Figure 11: Stomatal detections on a sorghum leaf using SAM (a and b) and our proposed (c and d) model. The blue regions depict the intersection of the ground truth, and the model prediction. The red shows the part of the ground truth which does not overlap with the model prediction, while the green detects the part of the predicted stomata that does not overlap with the ground truth.

similar trends in the results. For these datasets, we reported only the stomatal density and their centroid coordinates. To evaluate performance, we used MAE and MSE.

For costal zone detection, the evaluation was more straightforward, as the number and positions of costal zones were easily countable. SAM performed poorly in this task, as it relies on detecting relatively homogeneous micro-textures to define regions. Costal zones, however, are finely striated structures that SAM detects as multiple smaller regions. In addition, SAM's training on CV datasets makes it less suitable for microscopic imagery, further limiting its performance. Consequently, our model outperforms SAM by a substantial margin in detecting costal zones across all crop types.

Figure 11 shows the stomata detections using the SAM model. In the figure, the blue regions represent the intersection between the ground truth and the SAM predictions. The red regions indicate the part of the ground truth that does not overlap with the SAM prediction, while the green regions highlight the predicted stomata that do not overlap with the ground truth. Ideally, we aim for more blue and less red and green, which would indicate a higher degree of overlap between the predictions and the ground truth.

For the stomata predictions, it is observed that SAM either completely misses detecting a stomata. But if it does detect one, the boundary prediction is usually accurate. In contrast, our model successfully detects most of the stomata, though there is still room for improvement to achieve a perfect boundary match.

## 7.2 Ablation studies

In this section, we present ablation studies of the different components of the model and evaluate various experimental setups. These studies are detailed in the following subsections.

**- Without decision fusion:** Our proposed model utilizes three main data channels: i) intensity-based predictions, ii) topography-based predictions, and iii) disk detector-based predictions. We combine the results from these three data streams to make the final stomata cell detections. In this section, we perform ablation studies to evaluate the performance of each individual component. As shown in Table 4, intensity-based stomata predictions outperform the other two data channels by a significant margin. Although the topography channel appears to perform poorly at first glance, it is often able to detect certain stomata cells that are missed by the intensity-based model. This becomes particularly evident when examining the results from combinations of two model components. Furthermore, we demonstrate that combining all three data channels yields the best overall results.

Table 4: Ablation of network components, costal zone suppression (CZS) block, and active annotation iterations of the proposed model on the gold standard dataset of sorghum for the detection of stomata using Precision, recall and IoU.

Table 5: Stomata detection using the proposed model on sorghum (left). Stomata boundaries are marked in green. Right figure shows its corresponding ground truth. Blue depicts IC cells and red depicts annotated stomata.

| | Ablation experiments | Precision | Recall | IoU |
|---|---|---|---|---|
| Modality | Topography | 0.392 | 0.374 | 0.274 |
| | Intensity | 0.721 | 0.744 | 0.698 |
| | Disk detector | 0.634 | 0.825 | 0.554 |
| | Topography+intensity | 0.798 | 0.810 | 0.736 |
| | Topography+disk | 0.729 | 0.695 | 0.707 |
| | Disk detector+ intensity | 0.801 | 0.822 | 0.747 |
| | Disk detector+ intensity + topography | 0.844 | 0.893 | 0.782 |
| CZS | with suppression | 0.844 | 0.893 | 0.782 |
| | w/o supression | 0.649 | 0.0.728 | 0.696 |
| Feedback | No active feedback | 0.796 | 0.853 | 0.765 |
| | 10 active feedback | 0.832 | 0.875 | 0.780 |
| | 20 active feedback | 0.844 | 0.893 | 0.782 |
| | 30 active feedback | 0.849 | 0.899 | 0.783 |
| | 40 active feedback | 0.849 | 0.901 | 0.786 |

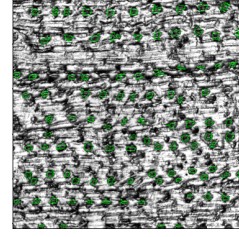 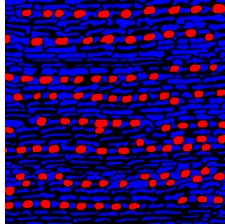

**- Without costal zone suppression:** Our proposed model includes a costal zone suppression branch. A costal zone is a region where stomata are typically absent. This block helps to suppress detections in the costal regions, which is particularly useful given the high confusion between prickles and stomata cells. Both structures share similar geometrical characteristics and textures. As shown in Table 4, the costal region suppression significantly reduces the number of false positives, as indicated by the noticeable improvement in the overall precision of stomata detections. We demonstrate experimentally that the costal zone suppression block effectively enhances the overall performance of stomata detection.

**- Active annotation:** Once we obtain the initial batch of results, we select 10 images with decently high IoU scores and recall values. These informative images are then used in the next annotation round and added to the training dataset. This process is repeated until the model performs consistently well across a wide variety of crop images. To evaluate the impact of active learning, we tested the results with no active feedback and with 10, 20, 30, and 40 active feedback images in batches of 10. Initially, without active feedback, we achieved an IoU score of 0.765. After

two rounds of active feedback (20 images), the IoU improved to 0.782, demonstrating the efficacy of this approach. However, as we increase the number of rounds, the performance improvement becomes marginal, showing only a 0.4% increase. As a result, we stopped active feedback for sorghum after 2 rounds. For setaria, we found that 3 rounds of active feedback were the most effective.

## 8 Conclusions, Generalizability, and future work

Conclusions: We have developed an AI-based model for improved model for phenotyping of SD, costal zone density, and bulliform cells for understanding plant structure and function, with application in crop genetics and biotechnology Lunn et al. (2024).Monitoring and understanding plant health plays a crucial role in improving crop yield. Our framework provides a generalized approach for detecting stomata, costal zones, and bulliform cell zones, even with minimal training data. By incorporating domain-specific plant knowledge, we further refine the detection results. Our model demonstrates that separating spatial structure knowledge from intrinsic cell models significantly reduces noise in the spatial distribution and structure of cells, enhancing the accuracy of our predictions. Additionally, we show that using multimodal data (intensity, topography, and region detectors) leads to the extraction of robust features, which in turn improves the model's overall performance. To address the challenge of limited annotated data, we propose leveraging active feedback control, which has proven effective in refining the model's predictions. Through extensive quantitative and qualitative experiments on three different crop types: setaria, sorghum, and maize demonstrate the efficacy of our method, achieving significant improvements over state-of-the-art segmentation techniques.

Generalizability: Detection and segmentation of objects and regions in images is a common problem in biological research and development, which remains a challenging bottleneck in data acquisitions. Similar to the problem of stomata detection, detecting mitochondria in electron microscopy images can be challenging because of the presence of other subcellular structures and imaging artifacts. The proposed method could be extended to these kind of problem statements too. While the three fundamental blocks would remain similar, the domain knowledge block would change, customized for the given dataset. A few more possible area of application of this method could be in Transmission-EM for the analysis of SARS-CoV-2 infected intestine organoids Lamers et al. (2020), 3D-imaging for cell biology and ecology of environmental microbial eukaryotes Colin et al. (2017), time-lapse microscopy for phenotypic profiling of the human genome Neumann et al. (2010).

Future work: As highlighted in Challenge 5, specular reflections often complicate the boundaries of various cell types. These reflections cause certain parts of the cells to appear brighter than others, altering the true nature of the object and making the segmentation more difficult. To address this challenge, future work could focus on mitigating the impact of specular reflections.

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
