# OpenReview forum: "Plant Phenotyping from Limited Training Data by Active Learning and Annotation"
_TMLR — Withdrawn by Authors_

### Review · Reviewer_cKFH · 2026-04-28

**Summary Of Contributions:**

**Summary**

This paper proposes a domain-knowledge-guided framework for plant phenotyping from limited annotated data. The method targets stomata detection and costal/bulliform zone segmentation in microscopic leaf epidermis images across three crop species: Setaria viridis, Sorghum bicolor, and Zea mays. The framework combines intensity and topography images, texture filters, geometric post-processing, evidence maximization, and crop-specific domain rules. It also includes an “active annotation” procedure where selected model outputs are added back into the training set. The paper reports improved performance over SAM on stomata detection and zone segmentation tasks, using precision, recall, IoU, MAE, and MSE as evaluation metrics.

**Strengths**

1. The topic is important and practically relevant. Automated plant phenotyping is a meaningful application area, and reducing the need for dense expert annotation could have practical value for plant biology and crop science.

2. The use of biological domain knowledge is well motivated. The paper makes a reasonable case that spatial regularities, cell geometry, topography, and crop-specific anatomical constraints can improve detection beyond generic segmentation models.

**Weaknesses**

1. My main concern is that the paper appears to be primarily an application paper in plant phenotyping, while the machine learning contribution is relatively limited for the TMLR audience. The core novelty lies more in domain-specific engineering and biological prior integration than in a broadly useful learning method.

2. The use of the term “active learning” is potentially misleading. In machine learning, active learning usually refers to an iterative process where the model /sampling strategy selects informative unlabeled samples for human annotation to improve the training set efficiently. In this paper, the process seems closer to manually selecting model predictions that already perform well and adding them to the training data. This is not the standard meaning of active learning and should be renamed or more carefully justified.

3. The paper claims a general cross-crop framework, but many parts of the method rely on crop-specific rules and design choices. Although the overall pipeline is shared, the need to customize the domain knowledge block and hyperparameters for each crop weakens the generality claim.

4. The experimental comparison is not sufficiently comprehensive. The main baseline is SAM with additional filtering rules. This does not fully establish the advantage of the proposed framework over other supervised, weakly supervised, few-shot, or domain-adapted segmentation/detection methods.

5. The presentation is sometimes difficult to follow. The background and biological motivation occupy substantial space, while the central methodological and evaluation points could be made more clearly and compactly.

**Audience:**

No

**Audience Explanation:**

Only to a limited extent. The application is important and the integration of domain knowledge is useful, but the machine learning contribution may not be sufficient or general for the broader TMLR community. The paper would likely be more naturally positioned for a plant phenotyping, computational biology, or applied computer vision venue unless the authors strengthen the methodological contribution and provide broader ML-oriented evidence.

**Claims And Evidence:**

No

**Claims Explanation:**

Partially. The experimental results support the claim that the proposed domain-knowledge-guided system can outperform SAM on the evaluated plant phenotyping tasks. However, the broader claims about active learning and general cross-crop applicability are less well supported. The “active learning” component does not align with the standard ML definition, and the method still appears to require substantial crop-specific design and hyperparameter choices.

**Requested Changes:**

1. Clarify the terminology around “active learning.” If the method does not use an acquisition function to select informative samples for annotation, it should probably be described as self-training with human validation, iterative annotation, or feedback-based dataset expansion rather than active learning.

2. Temper the generality claims. It would be more accurate to state that the framework is modular and can be adapted to new crops by modifying the domain-knowledge block, rather than claiming a fully general cross-crop method.

3. The experimental section should include stronger baselines beyond SAM, especially methods that are closer to the actual setting of limited annotated training data.

4. The organization and readability of the paper could be improved. In particular, the introduction and biological background could be shortened, and the method section could more clearly distinguish generic components from crop-specific components.

5. It would be better to provide a clearer account of how samples are selected during the iterative annotation process, how much human effort is required, and how this compares to standard active learning or manual annotation.

---

### Review · Reviewer_xErK · 2026-04-30

**Summary Of Contributions:**

This paper develops an automated segmentation technique for the phenotyping of epidermal cell patterns, specifically targeting costal and bulliform cell regions. The authors present an approach that integrates domain knowledge into a machine learning pipeline, starting from the most structured parts of the images and progressively extending the inference to regions with more ambiguous structures. Furthermore, the pipeline incorporates an active annotation strategy to continuously expand the training dataset throughout the learning process. The utility of the proposed method is demonstrated by applying two specific object and region detection tasks to images of three different crop species.

**Additional Comments:**

The paper provides detailed explanations regarding specific challenges in crop genetics and biotechnology, making it accessible to readers who are not familiar with the field.
From the perspective of a machine learning researcher, however, there are several major issues as follows:

- Lack of citations for fundamental machine learning literature: The paper fails to cite standard references in pattern recognition or representative works on core technologies such as active learning and CNNs. This omission raises doubts about the authors' understanding of machine learning techniques.

- Misinterpretation of "active learning": Although the title includes the term "active learning, " the actual technique employed appears to be self-taught learning or semi-supervised learning in a broad sense. This further suggests that the authors do not have a sufficient grasp of the relevant technologies.

- Technical unreliability: The formulation in Equation (1) is sloppy, which leaves the impression that the paper lacks technical rigor and reliability.

**Audience:**

No

**Audience Explanation:**

Unfortunately, this paper fails to demonstrate significant results within the context of machine learning. While the findings might be meaningful in the field of crop genetics and biotechnology, it is unlikely to contain content that would appeal to the TMLR audience.

**Claims And Evidence:**

Yes

**Claims Explanation:**

The authors claim that their proposed machine learning approach outperforms state-of-the-art segmentation methods in terms of both accuracy and time efficiency, and this claim appears to be supported by their experimental results.

**Requested Changes:**

This paper is entirely out of the scope of TMLR and is not written as a machine learning journal paper. Since it is unlikely that technical modifications will resolve this fundamental issue, no specific revisions are requested.

---

### Review · Reviewer_FiSh · 2026-05-21

**Summary Of Contributions:**

The paper proposes a framework for detecting stomata, costal zones, and bulliform zones from microscopy images with limited annotations. The framework takes both intensity and topography as input and processes them through four blocks. The authors incorporate domain-specific knowledge, separate spatial structure knowledge from intrinsic cell models, and use active learning to iteratively expand the set of manually annotated images.

**Audience:**

No

**Audience Explanation:**

The main focus of the paper is more on applying ML to plant phenomics and might be better suited to a non-ML venue. It does not provide a novel ML contribution or offer a pipeline that would be broadly useful to the TMLR audience.

**Claims And Evidence:**

No

**Claims Explanation:**

While the paper is well written, it is difficult for me, as a non-biologist, to understand. I can only assess the machine learning (ML) components, which are vague and lack details. The few ML parts are described at a high level, without architecture, model specifics, or hyperparameters, and the contribution is a pipeline built on domain knowledge that I can not accurately assess. The paper might be better suited to a non-ML venue, as it does not make a novel ML contribution or offer a pipeline that would be broadly useful to the TMLR audience.

**Requested Changes:**

Again, I do not believe the paper is well-suited for TMLR. For completeness, however, the requested changes I would suggest are to include more ML-specific details: architecture, loss function, optimizer, training hyperparameters, dataset splits, and more ablations isolating contributions.

---

### Note · Authors · 2026-05-29

**Comment:**

As suggested by the reviewers, we feel our paper is indeed suited for other avenues.

**Withdrawal Confirmation:**

I have read and agree with the venue's withdrawal policy on behalf of myself and my co-authors.